# Number of Children and Female Labor Participation in China

**DOI:** 10.3390/ijerph19148641

**Published:** 2022-07-15

**Authors:** Ke Wang, Guitao Zhang, Mengru Yu, Yangfei Gao, Yangyan Shi

**Affiliations:** 1School of Statistics, Shandong University of Finance and Economics, Jinan 250220, China; 20191724229@mail.sdufe.edu.cn; 2Business School, Qingdao University, Qingdao 266071, China; 3The Financial Department, University of International Business and Economics, Beijing 100029, China; 18375405338@163.com; 4School of Humanities, Creative Industries and Social Sciences, The University of Newcastle, Callaghan, NSW 2300, Australia; 5Macquarie Business School, Macquarie University, Sydney, NSW 2109, Australia

**Keywords:** number of children, female labor involvement, “inverted U-shaped”

## Abstract

The continuous decrease in the number of women of childbearing age and the consequent decrease in reproductive willingness have contributed to the continuous decrease in labor participation among Chinese women, which has negatively affected the stable socioeconomic development in terms of health. This paper deeply explores the intrinsic relationship between the number of children and women’s labor participation based on 2016 data from China Labor-force Dynamic Survey (CLDS). Our results show that there is an “inverted U-shaped” relationship between the number of children and the rate of women’s labor involvement; in other words, women’s labor participation shows a trend with the increase in the number of children, first rising and then falling; meanwhile, the relationship is more pronounced among women in eastern and central regions and towns. To this end, this study provides a theoretical research basis to effectively alleviate women’s selective pressure at home and work, and has a certain reference value for the Chinese government to improve women’s employment environment.

## 1. Introduction

Socioeconomic development is inseparable from effective labor supply, which in turn is jointly affected by the number of working-age population and the rate of labor participation. According to the data of the Seventh National Census, the total Chinese population reached 1,411,780 thousand, and the working-age population between 16 and 59 was 894,370 thousand. In recent years, with the increasing problem of population aging, the problem of population structure has been paid more and more attention. Although the total population of our country continues to grow slowly, and the rank of labor resources remains first place worldwide, at the same time, the proportion of our country’s population aged 65 years and older has reached 13.50%, and the elderly population ushered in a new total. The problem of population aging in China has continued to deepen (the data of population age structure comes from the National Bureau of Statistics) since it began to walk into an aging society in 2000 (according to the traditional standards of the United Nations, an area where the elderly over the age of 60 account for 10% of the total population or the elderly aged 65 (and older) account for 7% of the total population is regarded as entering an aging society), and China will enter the accelerated aging phase in the 14th Five-year Plan period with a decreasing proportion of the working-age population. The diminishing dominance of the demographic dividend will have a greater impact on the sustainable and stable development of our economic society, which is dependent on its long-term demographic dividend. Because the trend of aging is irreversible, the rate of labor participation has played a crucial role in providing effective labor.

Analyzing and studying the impact of the number of children on labor supply behavior and wage level of married women in my country is of great significance for evaluating the impact of declining fertility rates on the labor market, analyzing changes in women’s employment situation, and examining the dynamics of gender wage differences. At the same time, existing experience has proved that the female labor force participation rate and its influencing factors in China are different from other countries. One of the most important reasons for China ranking first for its workforce participation rate is that the women’s labor force participation rate is much higher than the world average. During the period when family planning policy was practiced in China, women’s family responsibilities were released to some extent by the limited number of children and the decrease in fertility rate, which allowed more women to pursue their own career development. Combined with women’s increased economic competence, sense of equality, and subject status after entering the labor market, this has made a major contribution to China’s economic development, which has had a positive impact on the increasing labor participation rate among women in China. However, against the background that China’s labor market has stepped into a negative growth era, the female labor-force participation rate has also decreased from 63.98% in 2010 to 60.57% in 2019 (the labor force participation rate data comes from the World Bank); although this is still higher than the global female labor force participation rate of 51.26% in 2019, the fact is that China’s labor force participation rate is decreasing year by year and the female labor participation rate is decreasing more rapidly than that of male labor. The data used in this study come from CLDS 2016, and the year is quite different from 2019, so the female labor force participation rate is relatively high. In addition, the statistical caliber of these data includes more urban data from 29 provinces in China. From the perspective of the accuracy of the empirical results, sparsely populated remote areas are excluded. Therefore, the average value of this indicator is higher than the statistical data in 2019. Starting with the direction of the increasing female labor participation rate is more beneficial to improve the effective labor supply when the male labor force participation rate is basically stable. Therefore, the strengthening employment policy was proposed in the 14th Five-year Plan to exploit the potential of labor supply by safeguarding and improving the employability of key groups to cope with a situation in which the total shortage and structural shortages of labor coexist in China. Women’s labor market potential can be mined by advocating gender equality and promoting equity in employment for both genders, which will also greatly enhance the growth of China’s economy. 

What factors would make the expected number of children affect the participation of Chinese women in the labor market? Various real-world analyses show that analyzing the factors that affect China’s female labor market is of great significance to China’s population structure improvement and social development. There are many factors affecting women’s labor participation. Fang et al. (2021) [1] pointed out that women’s labor participation rate will be affected by economic system transition, multi-generation family structure, family elderly care and gender identity. Moreover, in China’s traditional family concept of “men in charge of the outside of their family and women in charge of the inside of their family”, women assume more responsibilities to take care of the elderly and children, which also limits women’s labor participation to a certain extent. In addition, China’s fertility policy has been gradually adjusted in order to cope with the aging. The ‘two-child’ and ‘three-child’ policies have been gradually relaxed since 2016, which is expected to change the current situation of China’s declining fertility rate. However, the data show that China’s population fertility rate, which was only 8.50% (the fertility rate data come from the National Bureau of Statistics) in 2020, had been declining for four consecutive years from 2017 to 2020. The fertility peak caused by the “universal two-child” policy did not arrive as the National Health and Family Planning Commission predicted in 2017. More and more women choose to have fewer children or not to have children. The main reason for the low willingness of women of childbearing age to have children is that this will have a negative impact on their performance in the labor market (Zhu & Zhu, 2015) [2]. Therefore, the number of children may be an important factor affecting women’s labor participation.

According to the above-mentioned phenomenon, this paper produces the following thoughts: will the number of children have an impact on the female labor participation rate? If so, what is the extent of its impact? Is there heterogeneity? Specifically, why is there strong regional heterogeneity in the question between the number of children and the female labor force participation rate in China? What are the theoretical factors inherent in it? More importantly, what is the transmission mechanism and theoretical influence mechanism between the number of children and the female labor force participation rate studied in this paper? Additionally, does this theoretical transmission mechanism match our expectations or reality? This paper attempts to answer the above questions through empirical research, so as to provide a basis for the government to formulate fertility policy and tap the potential of labor supply.

In view of this, this paper uses the 2016 China Labor-force Dynamic Survey data (CLDS) to study the impact of the number of children on women’s labor participation in the new era of population policy development from an individual micro perspective. The results show that the relationship between the number of children and women’s labor participation presents an ‘inverted U-shaped’ relationship. Multiple robustness test results show that the conclusion mentioned above is robust. At the same time, the heterogeneity study found that the number of children and women’s labor participation showed an “inverted U-shaped” relationship, there was obvious regional and urban–rural heterogeneity, and the increase in the number of children showed a more significant “inverted U-pattern” relationship for women’s labor participation in the eastern and central regions and for urban women.

The structure of the rest paper is as follows: the second part is the literature review and research hypothesis; the third part includes data selection, variable description and model setting; The fourth part is the empirical analysis; the fifth part presents the conclusions and enlightenment.

## 2. Literature Review and Research Hypothesis

Under the background of the decline of China’s labor participation rate, many scholars explore the reasons for the decline of labor participation rate and women’s labor participation rate. With economic and social development, people’s ideological level has been continuously improved, modern society has paid more attention to women, and women’s status has been continuously improved. Increasingly, studies are also exploring the influence of the mechanism and relationship between the number of children and the female labor force participation rate in the family relationship. Cai & Wang (2004) [3] revealed the relationship between labor participation rate and unemployment rate, and they found that unemployment is an important reason for the decline of labor rate. In terms of female labor participation rate, Shen et al. (2012) [4] found that multi-generational family structure has a noticeable positive impact on female labor participation. In addition, the family elderly care rate will also reduce the female labor participation rate (Wu et al., 2017) [5]. In addition, Lu and Ge (2020) [6] analyzed the impact of various factors including system transition, income level and industrial structure adjustment on women’s labor participation. The research results show that economic system transition is an important factor affecting women’s labor participation rate in China.

In recent years, with the adjustment of fertility policy, more and more scholars began to pay attention to the number of children and women’s labor participation. However, due to the differences between estimation methods and data processing, it is not possible to obtain a unanimous conclusion. The impact of fertility rate on women’s labor supply behavior is one of the most important topics in the field of labor economics. Although there have been a lot of empirical studies conducted on women’s labor supply behavior in the United States and western developed countries, research on women’s labor supply behavior in developing countries is still very limited. The decline in the female fertility rate in my country may lead to an increase in the female labor force participation rate, change the female employment structure, and may also narrow the gender wage gap. According to the labor supply theory, reproductive behavior will affect women’s labor supply behavior. Angrist & Evans (1988) [7] and Alice & Masao (1992) [8] believe that the increase in the number of children will remarkably reduce the probability of women’s labor participation. Budig (2003) [9] used the event history analysis to analyze the data of the National Longitudinal Survey of Youth (NLSY) from 1979 to 1994, and found that pregnancy and the number of preschool children hindered unemployed women from entering the labor market. Bloom et al. (2009) [10] found that the negative impact of fertility on women’s labor participation rate has a long-term nature through the study on women’s labor participation rate and total fertility rate in 97 countries, and the fertility rate may also affect the labor supply of elderly women, rather than only in the period of childbearing age. For working women, Olena et al. (2016) [11] found that the number of children has a negative impact on women’s labor participation and their income in both developing and developed countries, and the “fertility cost” increases with the increase in the number of children. Mary (2016) [12] found that the more children women have, the lower their labor participation rate and the shorter their working hours. Many studies in China also show that the increase in the number of children has a remarkable negative effect on women’s labor participation. Wei Ning et al. (2013) [13] found that having more children will reduce the participation rate of non-agricultural employment of rural women. Zhang (2020) [14] and Zhang & Gu (2020) [14] also believe that fertility is one of the key factors leading to the depreciation of human capital of urban women, and fertility has a greater negative impact on the employment of urban women and highly educated women. This is because the concept of a matriarchal society is deeply rooted in the hearts of the people in China. It is always believed that in a family, the role played by women is often the most critical. It will play a more subtle and irreplaceable role in the construction and maintenance of a home. At the same time, traditional cultural factors also believe that women often use softness to overcome rigidity, and to a large extent play a firmer supporting role than men.

However, some scholars hold different views and believed that childbirth has no obvious direct impact on the labor participation of employed women (Cheng, 1999; Budig, 2003) [9,15]. For example, Cheng (1999) [15] found that there is no relationship between female labor participation and negative predictive fertility in Taiwan, China. In view of the current situation in China, Zhang (2011) [16] used the data of the China Health and Nutrition Survey to draw a similar conclusion, which means the increase in the number of children has no significant impact on whether rural married women participate in non-agricultural employment and on the wages of women in non-agricultural employment. The main difficulty in studying the effects of fertility on women’s labor supply behavior is that women’s reproductive decisions and labor supply decisions may be made at the same time and affect each other. Women who tend to participate in formal employment may also tend to have fewer children, and employment status may also affect subsequent reproductive behavior. In addition, there are other unobtainable parental characteristics such as women and their spouses, which affect both reproductive decision making and labor supply. Existing studies have used the instrumental variable method to solve the endogeneity problem in identification. Das et al. (2003) [17] and Ebenstein (2010) [18] used twins as an instrumental variable for the number of children.

To sum up, the study on the relation between the number of children and women’s labor participation has been not consistent with that at home and abroad. This provides better ideas and methods for this study: (1) The existing foreign research on this topic is often based on the common social background that couples independently determine the number of children. However, China has strictly implemented family planning for more than 30 years, which has changed the natural selection of fertility, and the social background of fertility is very different from that of western countries. Under the background of the continuous promotion of China’s “two-child” and “three-child” policies and the decline of women’s labor participation rate, this paper uses the 2016 China Labor-force Dynamic Survey data (CLDS) to try to answer the impact of the number of children on women’s labor participation. (2) Most scholars believe that it is linear for the impact of the number of children on women’s labor participation. This study takes the above factors into account and creatively increases the square of the number of children. Then, it analyzes the nonlinear relationship between the number of children and women’s labor participation. (3) Most of the existing literature focuses on the reasons for the decline of female labor participation rate and the relationship between the number of children and female labor supply. There is a lack of in-depth exploration on the micro impact mechanism of the number of children on female labor participation. This study takes into account the heterogeneity of the implementation of China’s fertility policy between urban and rural areas and different regions, investigates in detail the influence of the number of children on the labor participation of female groups with different characteristics, and explores the influence mechanism of the number of children on female labor participation.

Due to factors such as age, gender, educational development, income growth, and industrial structure, the issue of female labor force participation cannot be ignored, and its role in social and economic development is self-evident. For women, their labor participation in decision making is affected by marriage, family, childbirth and other factors, which cannot be ignored. Childbirth causes families to face a higher cost of living. Birth cost refers to the sum of various expenses required for a child from pregnancy, birth to training into a qualified labor force. Lv (2021) [19] divided the birth cost into three parts: direct cost, indirect cost and opportunity cost. Direct cost refers to the expenses directly spent on children, indirect cost refers to the expenses that are not directly spent on children in order to have children, while opportunity cost refers to the time-cost of the opportunity to increase income lost in order to have children. The influence that raising children has on women’s labor participation is not the sole influence, and there are income effects and substitution effects. At present, the reason for the existence of non-single-influence effects is closely related to the reform and development of China’s economic structure. With the development and progress of China’s social economy, people not only pursue material wealth, but also have a strong pursuit of spiritual wealth. Therefore, different regions and groups at different levels have different responses to the impact of income increase. The income effect means that in order to make up for the loss caused by childbirth, working women will take care of their children and work at the same time, while unemployed women will join the labor market; the substitution effect refers to the decline in women’s labor participation due to the large amount of time and energy they invest to take care of their children. In fact, the combined effect of the income effect and substitution effect is the decisive force that reflects the impact of the number of children on women’s labor participation. With the rapid development of China’s economy and the obvious improvement in people’s living standards, the cost of living brought about by childbirth has also increased drastically. As an important supporter of economic income in the family, the increasing economic pressure will encourage young women to carry out paid labor. Women have to choose employment, which improves women’s labor participation [20]. At this time, the increase in the number of children will significantly increase women’s labor participation.

However, compared with men, women need to bear more family responsibilities (such as taking care of the elderly, children, housework, etc.), consume more energy in the family, and have closer ties with the family. Therefore, although people are in the workplace, they may be gradually separated from the workplace in terms of psychology and behavior. When the increase in family responsibilities combined with the increase in the number of children is difficult to balance with labor participation, women may choose to sacrifice their labor participation time to engage in family production activities, and may even give up their career pursuit for family responsibility (Li, 2016) [21]. At this time, the increase in the number of children will obviously reduce women’s labor participation.

Based on the above analysis, within a certain range, the impact of the increase in the number of children on women’s labor participation mainly presents the income effect, that is, the increase in the number of children will significantly increase women’s labor participation. In other words, the effect of the number of children on the female labor force participation rate may not be a simple linear relationship. However, when the number of children continues to rise to a certain boundary value, the impact of the increase in the number of children on women’s labor participation mainly presents a substitution effect, that is, the increase in the number of children to a certain level will remarkably reduce women’s labor participation. In view of this, this paper puts forward the following research hypothesis:

**Research** **Hypothesis:** *there is an ‘inverted U-shaped’ relationship between the number of children and women’s labor participation*.

## 3. Data Source and Variable Description

### 3.1. Data Source

This paper uses the data of China Labor-force Dynamic Survey (CLDS) in 2016. CLDS is one of the “three major” construction projects of Sun Yat-sen University. The survey scope includes 29 provinces and cities in China (except Hong Kong, Macao, Taiwan, Tibet and Hainan). The database takes the lead in adopting the rotation sample tracking method in China, which can better adapt to China’s rapidly changing social and economic conditions. It has strong representativeness and provides a good data source for this study. This paper combines the individual questionnaire and family questionnaire in the CLDS in 2016, and selects the samples by the working status of female respondents in the observation year according to the questionnaire setting. A total of 3102 valid samples were obtained after excluding the missing samples of main variables such as labor participation and the number of children.

### 3.2. Data Selection

In terms of dependent variables, we consider that labor participation refers to the entire population of a certain age, with labor ability and employment requirements, engaged in a certain occupational labor. Referring to the research of Wei & Su (2013) [13], the question, “which of the following does your current work belong to?”, is used as the proxy variable of labor participation, and the respondents who answered that “currently working” and “currently not working” are assigned as 1 and 0 respectively. Therefore, the focus of this paper is on how the number of children affects whether women find work, rather than whether women are looking for work.

In terms of independent variables, take the question, “how many children have you had?”, as the proxy variable of the number of children, and refer to Yan’s (2020) [22] method to add the square term of the number of children to investigate its nonlinear impact on women’s labor participation.

In terms of control variables, based on the research of Zhang & Sun (2020) [23], three main control variables related to the research topic of this paper are selected, namely, female personal characteristics, work characteristics and family characteristics. Among them, personal characteristics mainly include age, education level, health status, urban and rural types, age of giving birth for the first time, etc., which are mostly related to young women’s labor participation; job characteristic variables mainly include employment industry, income, whether there is professional training, whether it is a full-time job, etc., and this paper attempts to measure workers’ work performance through job participation; family characteristics mainly select variables such as family annual income, family wealth, spouse labor participation, spouse education, spouse health status, whether the father is alive, and whether the mother is alive. Among them, the family wealth is measured by the variables extent of family wealth and family annual income. In particular, the extent of family wealth is an ordered discrete variable, and it is a variable that is positively measured: from 0 to 10, the higher the value, the higher the family wealth. It is a more objective and comprehensive measure to use the size of the numbers to measure the comprehensive wealth of the family more intuitively. Family annual income measures the material wealth of the family from the dimension of family income. With the continuous progress and development of social security, the employment situation of female labor force is also affected by social factors. As a typical measure of social security, insurance is gradually accepted and popularized by social groups. This paper selects two social security variables which are unemployment insurance and maternity insurance to measure the impact of social factors on female labor force participation. It is not purchased after the laborer’s career, but after the laborer decides to find employment. Buy this kind of insurance for yourself or your family at the time to truly achieve the purpose of unemployment benefits.

Considering the possible nonlinear influence of age on female labor participation, the square term of age is added to the regression based on Gu (2021) [24]. The main variable description and assignment in this paper are shown in Table 1.

The descriptive statistical results of the main variables are shown in Table 2. In the sample, 92.3% of women have jobs and 7.8% of women have no jobs, indicating that China has a high women’s labor participation rate and large scale of employment. This seems to be inconsistent with the value of the female labor force participation rate in 2019. However, from the perspective of data statistics, the data used in this study are from CLDS 2016, and the year has a large gap compared with 2019, so the female labor force participation rate is relatively high. In addition, the statistical caliber of these data includes more urban data from 29 provinces in China. From the perspective of the accuracy of the empirical results, sparsely populated remote areas are excluded. Therefore, the average value of this indicator is higher than the statistical data in 2019. In the sample, the average total household income of the previous year was CNY 42,531.49, accounting for 60.35% of those who thought their families were relatively rich (relatively rich: the extent of family wealth is >5). With China’s economic development and the increase in family income, residents with a higher family income will pay more attention to the quality of child rearing and reduce the number of children. Therefore, the change in income status will directly affect the family’s fertility decision making. Jia & Li (2019) [25] believed that there was a “U-shaped” relationship between family income and urban second child birth behavior, while Li et al. (2018) [26] found that there was an obvious negative relationship between family income and birth based on China’s current economic development. However, the fertility rate will improve when the economy develops to a certain extent and people no longer consider whether they can bear the economic pressure brought by raising multiple children. The education level of the sample is mainly junior high school education, of which 400 samples have reached the university education level or above, accounting for 12.89%. At the beginning of the founding of China, illiteracy accounted for 80% of the national population. Now the education level of China’s population, especially the education level of women, has significantly improved, which has significantly promoted women’s entry into the labor market. From our sample, women have an average of 1.76 children, of whom 2767 have 1–2 children, accounting for 89.2%, while 335 women have three children, accounting for 10.8%. The main reason for the absolute proportion of the sample with 1–2 children is that at the time of the survey, the implementation of family planning policy was having a remarkable impact on women’s reproductive choices in China. The greater the number of children, the greater the family care responsibilities and economic pressure women face. Elderly parents help their daughters take care of the housework wholeheartedly, which helps women invest more working time [4]. The average health status in the sample is 2.3, and it can be seen from the variable assignments in Table 1 that the value of the health status variable is measured inversely, that is, the smaller the value of the variable, the better the health status. Therefore, the mean value of health status in the sample is 2.3, and the per capita health status is considerable. With the improvement in living standards and the development of medical technology, the physical health of Chinese women is also improving.

### 3.3. Model Setting

This paper uses whether women have a job as a measure of labor participation, which is affected by the number of children. Since the Least Squares (OLS) is the most traditional and common method in regression, it is suitable for various types of variable regression. Therefore, this study firstly used OLS to perform regression analysis of the model. In this paper, the traditional OLS model is used for estimation, and the measurement model is set as follows:(1)$$FLP_{i}=\alpha_{0}+\beta_{0}X+\eta_{0}X^{2}+\gamma_{0}X_{1}+\tau_{0}X_{2}+\varphi_{0}X_{3}+\varepsilon_{i}$$
where $FLP_{i}$ represents the labor participation of young women, i is the sample, X represents the number of children, which is the core explanatory variable, and $X^{2}$ represents the square of the number of children; $X_{1}$ indicates personal characteristics of women; $X_{2}$ indicates the working status of women; $X_{3}$ indicates women’s family situation. $\beta_{0},\eta_{0},\gamma_{0},\tau_{0},\;\varphi_{0}$ indicate the parameters (to be estimated) for each variable, $\alpha_{0}$ is the constant term of the equation, and $\varepsilon_{i}$ represents the stochastic perturbation term of the equation. This paper judges the influence of the number of children on women’s labor participation decision making according to the regression result coefficient. If $\beta_{0}>0$, the number of children has no negative effect on women’s labor participation in decision making; If $\beta_{0}<0$, the number of children has a negative effect on women’s labor participation in decision making. If $\eta_{0}$ is significant, then the number of children has a nonlinear impact on labor participation decision making. $\varepsilon_{i}$ represents the stochastic perturbation term of the equation, and it is a random error term that represents any undetectable bias that can be missed in any model.

As the explained variable is female labor participation, which belongs to discrete dichotomous variable, in order to test the robustness of the impact of the number of children on female labor participation, this paper constructs probit discrete choice model of female labor participation. In addition, the probit model is suitable for variables whose type is ordinal and discrete. Therefore, this paper selects the ordinal probit model for benchmark regression and robustness test regression. The model set in this paper is as follows:(2)$$Pr\left(FLP_{i}\right)=f\left[\alpha +\beta X+\eta X^{2}+\gamma X_{1}+\tau X_{2}+\varphi X_{3}+\varepsilon_{i}\right]$$

Among them, β,η,γ,τ,φ represent the parameters to be estimated of each variable, respectively; α indicates the constant term of the equation, and $\varepsilon_{i}$ represents the stochastic perturbation term of the equation. According to the regression result coefficient β,η, we can judge the impact of the number of children on women’s labor participation in decision making. If β>0, the number of children has no negative effect on women’s labor participation in decision making; if β<0, the number of children has a negative effect on women’s labor participation in decision making. If β>0 and η<0, the relationship between the number of children and female labor rate is “inverted U”; if β<0 and η>0, the relationship between the number of children and female labor rate is “U-shaped”.

## 4. Empirical Analysis

In this paper, OLS and probit regressions are conducted separately, and the estimated results are shown in Table 3, with the first three columns being the OLS regression results and the last three columns being the probit regression results. Models (1) and (4) incorporate only the core explanatory variable number of children and squared number of children, while models (2), (3), (5) and (6) cumulatively add control variables and account for area fixed effects in turn to models (1) and (4) respectively. The reason for adding fixed effects is that the relationship between the number of children and the female labor force participation rate is largely affected by regional factors. For example, there are differences in the number of women and their employment development in the western and eastern regions, and because the economic development of the two regions is quite different, the research on the effect of regional factors on the female labor force participation rate cannot be ignored. Based on this analysis, this study added a regional fixed effect to the model regression. Specifically, this paper adopts the method of fixing each province to deal with fixed effects.

The regression results of the OLS model show that the number of children has a significant positive relationship with female labor force participation, with or without the addition of control variables and regional fixed effects, while the square of the number of children has a significant negative relationship with female labor force participation, i.e., there is a non-linear “Inverted U-pattern” relationship between the number of children and female labor force participation. This suggests that within a certain range, an increase in the number of children raised by women has an income effect on their labor force participation, while beyond this range, it has a substitution effect. The above results tentatively confirm the hypothesis of this study. Since the independent variable in this study is the number of children, the quadratic regression results of the number of children show a special phenomenon, which shows that the number of children and the female labor force participation rate are in an “inverted U” shape, which indicates that the number of children is closely related to the female labor force. Participation is not a simple linear relationship, so the follow-up research in this paper does not directly use linear variables for regression, which may lead to biased conclusions, but uses quadratic variables to verify conclusions and explain problems.

The regression results of the probit model show that the number of children has a significant positive effect on female labor force participation at the 1% level of significance; the square of the number of children has a significant negative effect on female labor force participation at the 5% level. With the inclusion of control variables, the squared number of children has a significant negative impact on female labor force participation with or without the addition of area fixed effects, and in both cases the relationship between the number of children and female labor force participation is non-linear and “inverted U-pattern”. This is consistent with the OLS model regression results, further validating the hypothesis of this study.

The results in Table 3 do not capture the extent to which the number of children affects female labor force participation; therefore, the paper further calculates the average marginal effect of each explanatory variable on female labor force participation and the estimated results are shown in Table 4. In order to test the stability of the effect of the number of children raised on female labor force participation, this paper uses a stepwise regression method to include explanatory variables in turn for estimation. Models (1)–(4) cumulatively incorporate, in turn, core explanatory variables (number of children, squared number of children), personal characteristics variables (age, education level, health status), job characteristics variables (log personal income, professional training, professional certification, type of workplace, whether or not full-time, maternity insurance, unemployment insurance) and other control variables (spouse’s education level, spouse’s labor force participation, spouse’s health status, father’s survivorship, mother’s survivorship, log of total household income, household affluence). The results of model (1) show that the effect of the number of children on the rate of female labor force participation still has a significant “inverted U-pattern” structure when area fixed effects are added. The sign of the number of children and the square of the number of children remain unchanged when the variables of personal, work and household characteristics are cumulatively included in turn. This is because having children increases the size of the family and the cost of living, which gives women an incentive to work to increase family income, but having too many children increases the amount of time women spend on family work and increases the market cost of childcare above women’s labor wages, forcing women to return to the home. Thus, the number of children remarkably influences women’s choice between family and work.

Regarding the effect of control variables on labor force participation, the regression results in Table 4 show that among the individual characteristics, age and age squared have a significant effect on female labor force participation, and there is an “inverted U-pattern” relationship between female age and labor force participation. This is due to the fact that younger women generally have higher educational qualifications than older women, and therefore, younger women have more employment options available. As women get older, family responsibilities such as marriage and childcare force more women to return to the home, while as children get older, women have more autonomy to choose between family and career. A positive coefficient on female educational attainment means that women with a good education are more likely to choose employment. This is because more educated women have more employment opportunities, and their entry into the workforce will result in higher salaries, thus providing an incentive for highly educated women to enter the labor market. The negative coefficient for health status may be explained by the fact that for both men and women, good health is always a basic requirement for people to choose employment or to continue their career.

In terms of job characteristics, personal income, type of workplace, whether or not they have maternity insurance, and whether or not they have unemployment insurance all significantly affect female labor force participation. In particular, the marginal effect of female personal income is positive, indicating that the higher the female labor income, the higher her labor force participation, as economic income in the previous year positively motivates female workers to actively participate in the market labor force. The marginal effect of workplace type is positive, showing that the more flexible the workplace type, the higher the female labor force participation. This is because a larger number of children may lead women to prefer jobs or occupations with fewer working hours and more flexibility, and the interruption of their work due to child-rearing may cause women to lose some of their more permanent jobs. The presence of maternity insurance and unemployment insurance has a significant positive impact on women’s participation in the workforce, as maternity insurance can prevent women from quitting before giving birth by providing maternity leave that protects their right to work, and unemployment insurance can protect the unemployed by paying unemployment benefits, so women workers may work harder to qualify for maternity insurance and unemployment insurance.

In terms of household characteristics, the level of education of the spouse, the labor force participation status of the spouse and whether the mother is alive are all significantly associated with women’s labor force participation decisions. In particular, the spouse’s education has a significant negative effect on women’s labor force participation decisions, suggesting that a higher spouse’s education level reduces the probability of women entering the labor market, possibly because men with higher education have higher wages and wealthier families, making women’s labor force participation demand lower. Spouse’s labor force participation status has a significant positive impact on women’s labor force participation decision. Women’s participation in the labor market increases by 0.51 when her spouse’s labor force participation increases by one unit, which is because with the development of the economy in modern society, women’s ideological and viewpoints are gradually breaking the shackles. They believe that women should not only have a husband and children, but should pursue higher social status and highlight their social values. The experience that men get at work is more attractive to women’s desire to work, so the above regression results appear. In addition, the presence or absence of a mother also significantly influences women’s labor force participation decisions, with the presence of a mother increasing women’s labor force participation by 0.22, reflecting the contribution of intergenerational grandparental care to married women’s labor force participation, which reduces the stress and mental strain on young women of caring for children and working at the same time. Since it is primarily women who are responsible for the care of children in the grandparents’ generation, the presence or absence of a father does not have a significant effect on women’s labor force participation decisions.

### 4.1. Robustness Tests

#### 4.1.1. Replacement Regression Model

This paper uses the method of replacing regression models for robustness testing. As the explanatory variables are binary discrete variables, this paper chooses a binary logit model to test the stability of the effect of the number of children on female labor force participation and constructs the following regression model.
(3)$$y_{i}=\alpha_{0}+\alpha_{1}X+\alpha_{2}X^{2}+\alpha_{3}X_{1}+\alpha_{4}X_{2}+\alpha_{5}X_{3}+\varepsilon_{i}$$
where $y_{i}$ denotes the ith female labor force participation status, X denotes the number of children and $X^{2}$ denotes the square of the number of children; $X_{1}$ is the female personal characteristics; $X_{2}$ denotes the female work characteristics and $X_{3}$ is the female family characteristics. $\alpha_{1},\alpha_{2},\alpha_{3},\alpha_{4},\alpha_{5}$ denote the parameters to be estimated for each variable respectively; $\alpha_{0}$ is the constant term of the equation and $\varepsilon_{i}$ is the random disturbance term.

The regression results are shown in Table 5. Model (1) includes only the core explanatory variables of number of children and squared number of children, model (2) includes all explanatory variables except for female household characteristics, and models (3) and (4) both include all explanatory variables, with model (4) adding regional fixed effects. In model (1), the number of children is significantly positively related to female labor force participation at the 1% level, and the square of the number of children is negatively related to female labor force participation at the 5% level, in line with the baseline regression results. Again, this shows the “inverted U” structure of female labor force participation rates increasing and then decreasing as the number of children increases. When all the control variables are added to model (4), the sign of the number of children and the square of the number of children does not change, but the significance of the square of the number of children increases. Among the control variables, personal income, type of workplace, whether or not one has unemployment insurance, spouse’s education level, spouse’s labor force participation status and whether or not the mother is alive have a significant positive effect on women’s labor force participation rate.

The regression results of model (4) are not significantly different from model (3) when regional fixed effects are added to the model. Thus, the regression results of the Logit model further illustrate the robustness of the original model, while once again validating the research hypothesis of this paper. The results obtained by applying the Logit model for regression are consistent with the benchmark regression results in this paper, which just verifies the accuracy of the research conclusions in this paper and is consistent with our expectations.

#### 4.1.2. Ending Data Processing

In order to remove the influence of the extreme values of the sample data on the estimation results, the number of children was subjected to 1% and 3% tailoring in this paper, and the model in the benchmark regression was re-adopted to empirically test the effect of the number of children on women’s labor force participation (please see Table 6).

#### 4.1.3. Endogeneity Test

Both the theoretical and empirical analyses above show that the number of children has a significant impact on female labor force participation. However, as women take into account the costs of childbearing, such as the possibility of diverting a significant amount of time and energy from work, or even losing their current job, they may have fewer children. In reality, therefore, there is an interaction between fertility and female labor force participation, leading to endogeneity problems in the model. This paper uses an instrumental variables approach to address the endogeneity issues mentioned above. In the analysis of the effect of age at childbearing on married women’s labor force participation, Yan (2020) [22] argues that delaying the age at childbearing significantly increases married women’s labor force participation. Therefore, this paper uses “age at first birth” as an instrumental variable to identify the number of children.

The results of the 2SLS regression using the instrumental variable “age at first birth” show that the first stage instrumental variable F-statistic is 52.71 (A rule of thumb for testing weak instrumental variables is that the F-statistic for the first stage regression should be greater than or equal to 10), making age at first birth a strong instrumental variable, and the sample passes the heteroskedasticity robust Hausman test, suggesting that the number of children and female labor force participation are endogenous.

Therefore the number of children for females can be expressed by Equation (4).
(4)$$X=\alpha_{0}+\alpha_{1}Z_{i}+\alpha_{2}F_{ij}+\mu_{i}$$
where X is the number of children, $F_{ij}$ denotes the individual characteristic variable mentioned in Equation (1) and $Z_{i}$ denotes the instrumental variable age at first birth. Equations (2) and (1) together form the baseline analytical model for IV. The assumptions identified by the model contain two conditions: first, age at first birth is highly correlated with the number of children ($\alpha_{i}\neq 0$). Second, age at first birth is not correlated with the error term ($\varepsilon_{i}$) in Equation (1).

Table 7 presents the results of the 2SLS and IV-probit models for estimating female labor force participation. Since the independent variable in this study is the number of children, the quadratic regression results of the number of children show a special phenomenon, which shows that the number of children and the female labor force participation rate are in an inverted “U” shape, which indicates that the number of children is closely related to the female labor force. Participation is not a simple linear relationship, so the follow-up research in this paper does not directly use linear variables for regression, which may lead to biased conclusions, but uses quadratic variables to verify conclusions and explain problems. The first two of these columns report the first stage and instrumental variable estimation results for IV-probit when no area fixed effects are added; the last three columns report the first stage and instrumental variable estimation results for IV-probit estimation when area fixed effects are added. As shown in model (5), the results show that after treating ‘age at first birth’ as an instrumental variable and adding regional fixed effects, having an additional child increases female labor force participation by 8.2%, but this is not significant, possibly due to the influence of the one-child policy and changes in fertility attitudes, with a larger sample of only children in the sample and an average number of children of 1.76, which affects the significance of the number of children coefficient. However, the number of children coefficient has a positive sign, which is consistent with the above regression results. This indicates that a certain increase in the number of children does not reduce female labor force participation. On the contrary, it will facilitate female labor market participation.

### 4.2. Heterogeneity Analysis

#### 4.2.1. Regional Heterogeneity

The degree of economic development and industrial structure varies from region to region, and the more dynamic a region’s economic development is, the more job opportunities it has; the employment rate in the East is higher than the central and western regions (eastern region: Beijing, Tianjin, Hebei, Shanghai, Jiangsu, Zhejiang, Fujian, Shandong, Guangdong, Hainan, Heilongjiang, Jilin, Liaoning; Central Region: Shanxi, Anhui, Jiangxi, Henan, Hubei, Hunan; Western Region: Sichuan, Guizhou, Yunnan, Tibet, Shaanxi, Gansu, Qinghai, Liaoning, Xinjiang, Chongqing, Guangxi, Inner Mongolia). Based on this objective, this paper analyses the heterogeneity of the impact of the number of children on female labor market participation from a regionally heterogeneous perspective. Table 8 shows the effect of the number of children on female labor force participation in different regions, with an “inverted U-pattern” relationship between the number of children and female labor force participation in the eastern and western regions of the country, consistent with the results of the baseline regression. The possible reasons for this are that the eastern and central regions of China are more economically developed, having higher wages and higher education for women, but at the same time, the price level is higher in the eastern and central regions than in the western regions, so when the number of children increases, families need paid labor for women to maintain their livelihoods; in addition, when the number of children reaches a certain level, women will devote more energy to their families or even withdraw from the labor market. In contrast, the number of children has no significant effect on female labor force participation in the western region, where the level of economic development is less advanced and per capita income is lower, forcing women to participate in the labor force in order to increase household income.

#### 4.2.2. Urban–Rural Heterogeneity

In the context of changing population policies and the dualistic urban–rural structure of the new era, employment opportunities in rural areas are generally smaller than in towns and cities, and the constraints of fertility policies implemented during the family planning period differed between urban and rural areas. Different levels of fertility exist in urban and rural areas, and the labor force participation of married women in urban and rural areas is clearly different. Based on this objective situation, this paper analyses the heterogeneity of the impact of the number of children on female labor market participation from the perspective of different village residence types. Table 9 shows the effect of the number of children on female labor force participation in both the urban and rural subsamples, and Table 10 shows the marginal effect. Models (1)–(4) and (7)–(10) are both regression results after adding the number of children and the square of the number of children, personal characteristics variables, work and family characteristics variables, and area fixed effects in four sequential cumulative steps under urban household status; models (5) and (6) and (11) and (12) are both regression results under rural household status, with area fixed effects added to models (6) and (12). In terms of urban–rural differences, there are large differences in the impact of the number of children on women’s labor force participation status across the different household samples.

The regression results in Table 10 show that the number of children of urban women in model (1) has a significant positive marginal effect on labor force participation, and the square of the number of children has a significant negative marginal effect on labor force participation of women. The significance of the non-linear effect of the number of children on urban women’s labor force participation increases when control variables such as personal characteristics, work characteristics and family characteristics are added. The results of the marginal effects of the factors on female labor force participation in Table 10 show that when the number of children of urban females is in the range 1–2.67, the probability of urban females entering the labor market increases significantly by 13% for each additional child. The labor force participation of urban females increases with the number of children. On the one hand, the cost of living is higher in urban areas compared to rural areas and the cost of raising children is higher. Women have to choose employment to provide more financial resources for their families in order to maintain a good quality of life and to provide quality living conditions and education for their children; on the other hand, urban women are more likely to enter the labor market due to their higher level of education and due to their own human capital depreciation. When urban women have more than 2.67 children, the probability of an urban woman entering the labor market with one more child is significantly reduced by 2%. After adding regional fixed effects, the effect of the number of children on female labor force participation increases to 16 per cent. Female labor force participation increases significantly when urban women have more personal income, hold professional certificates, work full-time and have unemployment insurance, when urban women’s spouses enter the labor market, and when urban women’s mothers are alive. The estimates suggest that the marginal effect of the number of children on rural women’s labor force participation is not significant, probably because rural women generally have lower educational attainment and are limited in their employment options, and because for rural women, the traditional concept of “men in charge of the outside of their family and women in charge of the inside of their family” is deeply rooted, with more women choosing to take on the majority of household responsibilities and being less financially independent.

## 5. Conclusions

Based on the data from the 2016 China Labor-Force Dynamic Survey, this paper explores the effect of an increase in the number of children on female labor force participation. The results of this study show the following: (1) The relationship between the number of children and female labor and participation rates is not a simple linear relationship, but a nonlinear relationship with an inflection point. An increase in the number of children has a robust “inverted U-pattern” relationship with female labor force participation, which holds true after controlling for female personal characteristics, work characteristics, family characteristics and the addition of provincial fixed effects. (2) Further research found that higher prior-year earnings, unemployment insurance, and spousal labor status also significantly increased the likelihood of female labor force participation in China. (3) With the development of economy and society, the impact of economic structure and development speed on the female labor force participation rate cannot be ignored, and external factors such as regions will also affect the conclusions of this paper. In terms of heterogeneity, an increase in the number of children has a more significant “inverted U-pattern” relationship with female labor force participation in eastern and central China, and an increase in the number of children has a more significant “inverted U-pattern” relationship with female labor force participation in urban areas than in rural areas. In addition, this paper also focuses on providing policy suggestions and enlightenment for the conclusions of heterogeneity analysis. The existence of provincial fixed effects can also explain the influence of regions on the relationship between the number of children and the female labor force participation rate.

In view of this, the following insights have been drawn from this paper: (1) The government should more actively promote the concept of gender equality, raise women’s awareness of independence, enact relevant laws to guarantee women’s fair employment, further increase women’s labor force participation and liberate women’s economic potential to promote the further development of China’s economy. Raising women’s labor force participation might offset ongoing declines in male participation, and doing that will require support for women who have relatively large numbers of children. (2) The government should strengthen laws to reduce the level of labor force discrimination, increase employment opportunities for women, provide various forms of vocational skills training, and make efforts to improve the employment environment for women; steps should also be taken to support and guide the development of a social childcare service system, improve the level of protection, reduce the pressure of childcare on families with many children, and make efforts to increase women’s willingness to have children. (3) According to the research results of regional heterogeneity in this paper, it can be seen that in order to improve the female labor force participation rate and rationalize the social demographic structure, it is necessary to take measures from the perspective of heterogeneity. As China’s regional economic development is uneven and the dualistic economic and social structure between urban and rural areas will continue to exist for a long time, the effect of a universal two-child policy will vary greatly depending on the education level, income level and fertility attitudes of residents. Therefore, the government should make great efforts to universalize compulsory education and expand higher education, so as to increase the educational attainment of rural women and women in the western region, and to reduce the restrictions on the labor participation of married women in the rural and western regions due to the traditional social concept of “men in charge of the outside of their family and women in charge of the inside of their family”, so as to encourage women to leave the private sphere of the family, participate in employment and receive remuneration. Thus, this will lead to women moving out of the private sphere of the home and into gainful employment, thereby improving their economic status.

## Figures and Tables

**Table 1 ijerph-19-08641-t001:** Variable Description and Assignment.

	Variable Name	Name in Regression	Variable Assignment
dependent variable	Labor force participation of women	Participation rate	working = 1; not working = 0
independent variables	Number of children	Child num	
	Square of the number of children	Child num2	unit: piece
Control Variables	Age of women	Age	age (year)
	Educational attainment	Education	1 = Primary school education and below; 2 = Junior high school education; 3 = high school degree; 4 = college degree and above
	Health status	Health	Absolutely healthy = 1; very healthy = 2; Relatively healthy = 3; average = 4; unhealthy = 5
	Personal income	Ln income	Logarithm of personal income
	professional training	Train	attend = 1; not attend = 0
	Work industry	Industry	primary industry = 1; secondary industry = 2; tertiary industry = 3
	Work unit	Company	Civil servant or public institution = 1; State-owned enterprise or collective = 2; Foreign capital or private enterprise = 3; Individual or other = 4
	Whether it is a full-time job	Full-time	yes = 1; no = 0
	Whether to purchase maternity insurance	Maternity Insurance	yes = 1; no = 0
	Whether to purchase unemployment insurance	Unemployment insurance	yes = 1; no = 0
	Education attainment of spouse	Hus education	1 = Primary school education and below; 2 = Junior high school education; 3 = High school degree; 4 = College degree and above
	Health status of spouse	Hus health	Absolutely healthy = 1; very healthy = 2; relatively healthy = 3; average = 4; unhealthy = 5
	Labor participation of spouse	Hus work	working = 1; not working = 0
	Whether the father is alive	Father	yes = 1; no = 0
	Whether the mother is alive	Mother	yes = 1; no = 0
	Extent of family wealth	Rich	Getting richer from 0 to 10
	Family annual income	Ln fincome	Logarithm of family income last year

**Table 2 ijerph-19-08641-t002:** Descriptive statistical results of variables.

Name of Variables	Observed Value	Mean	Standard Deviation	Minimum	Maximum
Child num	3102	1.764	0.821	1	7
Child num2	3102	3.784	3.898	1	49
Participation rate	3102	0.926	0.262	0	1
Old	3102	40.762	4.705	20	51
Old2	3102	1716.392	576.770	400	2601
Education	3102	1.986	1.014	1	4
Health	3102	2.318	0.927	1	5
Ln income	3102	9.134	2.554	0	14.509
Train	3102	0.102	0.303	0	1
Unit	3102	3.456	0.870	1	4
Full-time	3102	0.854	0.353	0	1
Maternity insurance	3102	0.168	0.374	0	1
Unemployment insurance	3102	0.172	0.377	0	1
Hus education	3102	2.170	0.946	1	4
Hus work	3102	0.932	0.251	0	1
Hus health	3088	2.112	0.910	1	5
Father	3102	0.610	0.488	0	1
Mother	3102	0.748	0.434	0	1
Ln fincome	3101	10.658	1.008	6.217	14.078
rich	3101	6.203	11.239	1	10

**Table 3 ijerph-19-08641-t003:** Baseline regression.

Variables	OLS			Probit		
	(1)	(2)	(3)	(4)	(5)	(6)
Child num	0.06 ***	0.07 ***	0.07 ***	0.39 ***	0.43 ***	0.45 ***
	(0.02)	(0.02)	(0.02)	(0.12)	(0.13)	(0.13)
Child num2	−0.01 ***	−0.01 ***	−0.01 ***	−0.06 **	−0.07 ***	−0.07 ***
	(0.00)	(0.00)	(0.00)	(0.02)	(0.03)	(0.03)
Old2		0.00 ***	0.00 ***		0.00 ***	0.00 ***
		(0.00)	(0.00)		(0.00)	(0.00)
Education		0.00	−0.00		−0.00	−0.02
		(0.01)	(0.01)		(0.06)	(0.06)
Health		−0.01	−0.01		−0.07	−0.07
		(0.01)	(0.01)		(0.04)	(0.05)
Ln income		0.01 ***	0.01 ***		0.05 ***	0.05 ***
		(0.00)	(0.00)		(0.01)	(0.01)
Train		0.02	0.01		0.09	0.06
		(0.02)	(0.02)		(0.14)	(0.14)
Unit		0.03 ***	0.03 ***		0.21 ***	0.19 ***
		(0.01)	(0.01)		(0.05)	(0.05)
Full-time		0.00	−0.00		0.04	0.04
		(0.01)	(0.01)		(0.10)	(0.11)
Maternity insurance		0.03	0.04		0.27	0.34 *
		(0.03)	(0.03)		(0.20)	(0.20)
Unemployment		0.10 ***	0.10 ***		0.83 ***	0.77 ***
		(0.03)	(0.03)		(0.20)	(0.21)
Hus education		−0.01 **	−0.01 **		−0.10 *	−0.11 *
		(0.01)	(0.01)		(0.06)	(0.06)
Hus work		0.10 ***	0.09 ***		0.59 ***	0.51 ***
		(0.02)	(0.02)		(0.12)	(0.13)
Hus health		0.00	0.01		0.03	0.04
		(0.01)	(0.01)		(0.05)	(0.05)
Father		−0.01	−0.01		−0.05	−0.06
		(0.01)	(0.01)		(0.09)	(0.09)
Mother		0.02 **	0.03 **		0.20 **	0.22 **
		(0.01)	(0.01)		(0.09)	(0.09)
Ln fincome		−0.01*	−0.00		−0.07	−0.00
		(0.01)	(0.01)		(0.04)	(0.05)
Rich		0.00	−0.00		0.00	−0.00
		(0.00)	(0.00)		(0.01)	(0.00)
Cons	0.87 ***	0.58 ***	0.50 ***	0.99 ***	−0.68	−1.29 *
	(0.05)	(0.08)	(0.09)	(0.12)	(0.57)	(0.74)
Regional fixed effect	YES	NO	YES	YES	NO	YES
*N*	3102	3086	3086	3102	3086	3070
*R* ^2^	0.023	0.062	0.077			

Note: This section describes the significance of the regression results in the table. Standard deviations are in brackets, *** indicates significant at the 1% level, ** indicates significant at the 5% level, and * indicates significant at the 10% level. The following table is the same as above.

**Table 4 ijerph-19-08641-t004:** Marginal effects of probit regression.

Variables	(1)	(2)	(3)	(4)
Child num	0.42 ***	0.36 ***	0.47 ***	0.45 ***
	(0.12)	(0.12)	(0.13)	(0.13)
Child num2	−0.06 **	−0.06 **	−0.08 ***	−0.07 ***
	(0.03)	(0.03)	(0.03)	(0.03)
Old		0.16 ***	0.17 ***	0.16 ***
		(0.12)	(0.12)	(0.14)
Old2		−0.0025 ***	−0.002 ***	−0.002 ***
		(0.00)	(0.00)	(0.00)
Education		0.01	−0.08	−0.02
		(0.04)	(0.05)	(0.06)
Health		−0.06	−0.05	−0.07
		(0.04)	(0.04)	(0.05)
Ln income			0.04 ***	0.05 ***
			(0.01)	(0.01)
Train			0.06	0.06
			(0.14)	(0.15)
Unit			0.20 ***	0.19 ***
			(0.05)	(0.05)
Full-time			0.05	0.04
			(0.11)	(0.11)
Maternity insurance			0.35 *	0.34 *
			(0.20)	(0.20)
Unemployment			0.76 ***	0.77 ***
			(0.20)	(0.21)
Hus education				−0.11 *
				(0.06)
Hus work				0.51 ***
				(0.13)
Hus health				0.04
				(0.05)
Father				−0.06
				(0.09)
Mother				0.22 **
				(0.09)
Ln fincome				−0.00
				(0.05)
Rich				−0.00
				(0.00)
Cons	1.08 ***	0.55 **	−0.65	−1.29 *
	(0.37)	(0.22)	(0.54)	(0.74)
Regional fixed effect	YES	YES	YES	YES
*N*	3086	3102	3086	3070

* <0.1, ** <0.05, *** <0.001.

**Table 5 ijerph-19-08641-t005:** Robustness: replacement regression models (logit regression).

Variables	(1)	(2)	(3)	(4)
Child num	0.84 ***	0.92 ***	0.82 ***	0.86 ***
	(0.23)	(0.25)	(0.24)	(0.25)
Child num2	−0.12 **	−0.14 ***	−0.13 ***	−0.14 ***
	(0.05)	(0.05)	(0.05)	(0.05)
Old2		0.00 ***	0.00 ***	0.00 ***
		(0.00)	(0.00)	(0.00)
Education		−0.14	0.02	−0.02
		(0.10)	(0.11)	(0.12)
Health		−0.10	−0.12	−0.13
		(0.08)	(0.09)	(0.09)
Ln income		0.08 ***	0.09 ***	0.09 ***
		(0.02)	(0.02)	(0.02)
Train		0.15	0.18	0.11
		(0.28)	(0.28)	(0.29)
Unit		0.39 ***	0.41 ***	0.37 ***
		(0.10)	(0.09)	(0.10)
Full-time		0.06	0.06	0.03
		(0.21)	(0.21)	(0.22)
Maternity insurance		0.58	0.50	0.62
		(0.42)	(0.42)	(0.43)
Unemployment insurance		1.72 ***	1.82 ***	1.70 ***
		(0.45)	(0.45)	(0.46)
Hus education			−0.22 **	−0.22 **
			(0.11)	(0.11)
Hus work			1.13 ***	0.95 ***
			(0.23)	(0.23)
Hus health			0.05	0.06
			(0.09)	(0.10)
Father			−0.11	−0.13
			(0.18)	(0.18)
Mother			0.41 **	0.44 **
			(0.19)	(0.19)
Ln fincome			−0.12	0.00
			(0.09)	(0.09)
Rich			0.00	−0.00
			(0.01)	(0.01)
Cons	1.79 **	−1.68	−1.63	−2.82 *
	(0.76)	(1.04)	(1.14)	(1.44)
Regional fixed effect	YES	YES	NO	YES
*N*	3086	3086	3086	3070

* <0.1, ** <0.05, *** <0.001.

**Table 6 ijerph-19-08641-t006:** Robustness regression results after tailing treatment.

Variables	1% Tail Reduction		3% Tail Reduction	
	OLS	Probit	OLS	Probit
Child num	0.07 ***	0.54 **	0.07 ***	0.55 **
	(0.03)	(0.22)	(0.03)	(0.22)
Child num2	−0.01 **	−0.10 **	−0.01 **	−0.10 **
	(0.01)	(0.05)	(0.01)	(0.05)
Old2	0.00 ***	0.00 ***	0.00 ***	0.00 ***
	(0.00)	(0.00)	(0.00)	(0.00)
Education	−0.00	−0.02	−0.00	−0.02
	(0.01)	(0.06)	(0.01)	(0.06)
Health	−0.01	−0.07	−0.01	−0.07
	(0.01)	(0.05)	(0.01)	(0.05)
Ln income	0.01 ***	0.05 ***	0.01 ***	0.05 ***
	(0.00)	(0.01)	(0.00)	(0.01)
Train	0.01	0.06	0.01	0.06
	(0.02)	(0.15)	(0.02)	(0.15)
Unit	0.03 ***	0.20 ***	0.03 ***	0.20 ***
	(0.01)	(0.05)	(0.01)	(0.05)
Full-time	−0.00	0.03	−0.00	0.03
	(0.01)	(0.11)	(0.01)	(0.11)
Maternity insurance	0.04	0.34 *	0.04	0.34 *
	(0.03)	(0.20)	(0.03)	(0.20)
Unemployment insurance	0.10 ***	0.77 ***	0.10 ***	0.77 ***
	(0.03)	(0.21)	(0.03)	(0.21)
Hus education	−0.01 *	−0.10 *	−0.01 **	−0.10 *
	(0.01)	(0.06)	(0.01)	(0.06)
Hus work	0.09 ***	0.52 ***	0.09 ***	0.51 ***
	(0.02)	(0.12)	(0.02)	(0.12)
Hus health	0.01	0.04	0.01	0.04
	(0.01)	(0.05)	(0.01)	(0.05)
Father	−0.01	−0.05	−0.01	−0.06
	(0.01)	(0.09)	(0.01)	(0.09)
Mother	0.03 **	0.22 **	0.03 **	0.22 **
	(0.01)	(0.09)	(0.01)	(0.09)
Ln fincome	0.00	0.03	0.00	0.04
	(0.01)	(0.05)	(0.01)	(0.05)
Rich	−0.01	−0.04	−0.01 *	−0.04
	(0.00)	(0.03)	(0.00)	(0.03)
Cons	0.49 ***	−1.52 **	0.48 ***	−1.62 **
	(0.10)	(0.77)	(0.10)	(0.79)
Regional fixed effect	YES	YES	YES	YES
*N*	3086	3070	3086	3070
*R* ^2^	0.076		0.075	

* <0.1, ** <0.05, *** <0.001.

**Table 7 ijerph-19-08641-t007:** Endogeneity test.

Variables	(1)	(2)	(3)	(4)	(5)
	First Stage	Ivprobit	First Stage	2SLS	Ivprobit
Child num	−0.030 ***		−0.029 ***		
	(0.004)		(0.004)		
Child num2		0.097 *		0.090	0.082
		(0.409)		(0.050)	(0.442)
Old	0.086 ***	0.293 ***	0.085 ***	0.050 ***	0.309 ***
	(0.020)	(0.056)	(0.019)	(0.008)	(0.058)
Old2	−0.001 ***	0.003 ***	−0.001 ***	0.001 ***	0.004 ***
	(0.000)	(0.001)	(0.000)	(0.000)	(0.001)
Education	−0.118 ***	−0.038	−0.112 ***	−0.008	−0.052
	(0.022)	(0.085)	(0.022)	(0.010)	(0.089)
Health	0.059 ***	−0.060	0.035 **	−0.005	−0.058
	(0.015)	(0.049)	(0.015)	(0.006)	(0.048)
Ln income	−0.023 ***	0.045 ***	−0.018 ***	0.007 ***	0.041 ***
	(0.006)	(0.016)	(0.005)	(0.002)	(0.016)
Train	−0.097 *	0.051	−0.078	0.003	0.021
	(0.053)	(0.158)	(0.051)	(0.019)	(0.161)
Unit	−0.181 ***	0.030	−0.158 ***	−0.006	0.017
	(0.039)	(0.128)	(0.038)	(0.016)	(0.131)
Full-time	0.036	0.188	−0.010	0.032	0.292
	(0.076)	(0.219)	(0.074)	(0.026)	(0.225)
Maternity insurance	−0.198 ***	0.662 ***	−0.194 ***	0.067 **	0.622 **
	(0.075)	(0.242)	(0.073)	(0.028)	(0.247)
Unemployment insurance	−0.089 ***	−0.121 *	−0.086 ***	−0.018 **	−0.131 *
	(0.021)	(0.073)	(0.020)	(0.009)	(0.074)
Hus education	0.096	0.932 ***	0.140 *	0.190 ***	0.867 ***
	(0.085)	(0.173)	(0.082)	(0.030)	(0.183)
Hus work	−0.001	−0.002	−0.001	−0.000	−0.002
	(0.001)	(0.002)	(0.001)	(0.000)	(0.002)
Hus health	0.053 *	−0.146	0.059 **	−0.016	−0.167 *
	(0.031)	(0.095)	(0.030)	(0.011)	(0.099)
Father	−0.094 ***	0.225 **	−0.077 **	0.027 **	0.248 **
	(0.034)	(0.104)	(0.033)	(0.012)	(0.105)
Mother	−0.061 ***	−0.084	−0.048 ***	−0.002	−0.012
	(0.016)	(0.052)	(0.016)	(0.006)	(0.054)
Rich	−0.001	0.002	0.000	0.000	−0.000
	(0.001)	(0.006)	(0.001)	(0.000)	(0.005)
Cons	2.044 ***	4.683 ***	1.581 ***	−0.337 **	5.992 ***
	(0.415)	(1.251)	(0.431)	(0.168)	(1.319)
Regional fixed effect	NO	NO	YES	YES	YES
*N*	2853	2853	2853	2853	2821
*Wald chi2*	227.52 **		287.02 **		
*First-stage F*	52.30 ***		52.71 ***		

* <0.1, ** <0.05, *** <0.001.

**Table 8 ijerph-19-08641-t008:** Heterogeneity test: regional heterogeneity.

Variables	Eastern	Central	Western
Child num	0.40 **	0.66 ***	0.02
	(0.17)	(0.25)	(0.54)
Child num2	−0.07 **	−0.10 **	0.03
	(0.04)	(0.04)	(0.13)
O ld2	0.00 ***	0.00 ***	0.00 ***
	(0.00)	(0.00)	(0.00)
Education	−0.07	0.22 *	0.01
	(0.08)	(0.13)	(0.16)
Health	−0.12 *	−0.05	−0.11
	(0.06)	(0.09)	(0.11)
Ln income	0.07 ***	0.07 ***	−0.01
	(0.02)	(0.02)	(0.04)
Train	0.07	−0.07	0.19
	(0.18)	(0.33)	(0.43)
Unit	0.15 **	0.49 ***	0.16
	(0.07)	(0.11)	(0.13)
Full-time	0.02	0.35 *	−0.15
	(0.16)	(0.18)	(0.25)
Maternity insurance	0.13	0.77	0.00
	(0.23)	(0.57)	(.)
Unemployment insurance	0.83 ***	1.13 *	0.34
	(0.24)	(0.60)	(0.54)
Hus education	−0.13 *	−0.07	−0.20
	(0.07)	(0.12)	(0.16)
Hus work	0.46 ***	0.64 **	0.97 ***
	(0.16)	(0.29)	(0.31)
Hus health	0.06	0.03	−0.03
	(0.07)	(0.09)	(0.11)
Father	−0.17	0.09	0.08
	(0.12)	(0.17)	(0.21)
Mother	0.28 **	0.13	0.22
	(0.13)	(0.19)	(0.22)
Ln fincome	−0.03	0.02	−0.12
	(0.06)	(0.09)	(0.11)
Rich	0.00	0.01	−0.08
	(0.01)	(0.05)	(0.07)
Cons	−0.46	3.90 ***	0.99
	(0.82)	(1.17)	(1.54)
*N*	1536	1029	461

* <0.1, ** <0.05, *** <0.001.

**Table 9 ijerph-19-08641-t009:** Heterogeneity test: urban–rural heterogeneity.

Variables	Urban				Rural	
	(1)	(2)	(3)	(4)	(5)	(6)
Child num	0.72 **	1.16 ***	1.14 ***	1.38 ***	0.21	0.25
	(0.31)	(0.35)	(0.33)	(0.36)	(0.16)	(0.17)
Child num2	−0.15 *	−0.20 **	−0.22 **	−0.25 **	−0.05	−0.06 *
	(0.09)	(0.10)	(0.09)	(0.10)	(0.03)	(0.03)
Old2		−0.00 *	−0.00	−0.00	0.00 ***	0.00 ***
		(0.00)	(0.00)	(0.00)	(0.00)	(0.00)
Education		−0.04	0.07	0.05	−0.05	−0.05
		(0.09)	(0.10)	(0.11)	(0.08)	(0.08)
Health		−0.13	−0.09	−0.12	−0.08	−0.08
		(0.09)	(0.09)	(0.10)	(0.05)	(0.06)
Ln income		0.09 ***	0.09 ***	0.10 ***	0.05 ***	0.04 ***
		(0.03)	(0.03)	(0.03)	(0.02)	(0.02)
Train		0.02	0.11	0.05	0.08	0.10
		(0.21)	(0.20)	(0.22)	(0.22)	(0.23)
Unit		0.02	−0.00	−0.02	0.44 ***	0.43 ***
		(0.08)	(0.07)	(0.08)	(0.07)	(0.08)
Full-time		0.66 ***	0.51 **	0.70 ***	−0.02	−0.11
		(0.23)	(0.21)	(0.24)	(0.12)	(0.14)
Maternity insurance		0.89 ***	0.65 **	1.01 ***	−0.52 *	−0.52
		(0.31)	(0.28)	(0.32)	(0.31)	(0.33)
Unemployment insurance		0.46 *	0.50 *	0.48 *	1.29 ***	1.32 ***
		(0.28)	(0.26)	(0.29)	(0.36)	(0.38)
Hus education			−0.08	−0.13	0.19 ***	0.21 ***
			(0.10)	(0.11)	(0.07)	(0.08)
Hus work			0.48 **	0.41 *	0.65 ***	0.62 ***
			(0.20)	(0.22)	(0.17)	(0.17)
Hus health			−0.09	−0.14	0.08	0.09
			(0.09)	(0.10)	(0.06)	(0.06)
Father			−0.25	−0.29 *	0.07	0.02
			(0.16)	(0.17)	(0.11)	(0.12)
Mother			0.50 ***	0.63 ***	0.12	0.14
			(0.17)	(0.19)	(0.12)	(0.12)
Ln fincome			−0.11	−0.10	−0.05	−0.01
			(0.10)	(0.11)	(0.05)	(0.06)
Rich			−0.04	−0.09	0.00	−0.00
			(0.05)	(0.06)	(0.01)	(0.01)
Cons	0.98 *	−0.75	0.26	−0.01	−1.41 *	−1.53
	(0.54)	(0.91)	(1.08)	(1.36)	(0.73)	(1.02)
Regional fixed effect	YES	YES	NO	YES	NO	YES
*N*	925	925	1052	921	2034	2034

* <0.1, ** <0.05, *** <0.001.

**Table 10 ijerph-19-08641-t010:** Heterogeneity test: urban–rural heterogeneity (marginal effects).

Variables	Urban				Rural	
	(7)	(8)	(9)	(10)	(11)	(12)
Child num	0.11 **	0.14 ***	0.13 ***	0.16 ***	0.04 *	0.03
	(0.05)	(0.04)	(0.04)	(0.04)	(0.02)	(0.02)
Child num2	−0.02 *	−0.02 **	−0.02 **	−0.03 **	−0.01 *	−0.01 *
	(0.01)	(0.01)	(0.01)	(0.01)	(0.01)	(0.00)
Old2		−0.00 *	−0.00	−0.00	0.00 ***	0.00 ***
		(0.00)	(0.00)	(0.00)	(0.00)	(0.00)
Education		−0.00	0.01	0.01	−0.01	−0.01
		(0.01)	(0.01)	(0.01)	(0.06)	(0.01)
Health		−0.02	−0.01	−0.01	−0.04	−0.01
		(0.01)	(0.01)	(0.01)	(0.02)	(0.01)
Ln income		0.01 ***	0.01 ***	0.01 ***	0.00 **	0.00 ***
		(0.00)	(0.00)	(0.00)	(0.00)	(0.00)
Train		0.00	0.01	0.01	0.01	0.01
		(0.03)	(0.02)	(0.03)	(0.03)	(0.03)
Unit		0.00	−0.00	−0.00	−0.16	0.05 ***
		(0.01)	(0.01)	(0.01)	(0.01)	(0.01)
Full-time		0.08 ***	0.06 **	0.08 ***	−0.01	−0.01
		(0.03)	(0.02)	(0.03)	(0.02)	(0.02)
Maternity insurance		0.11 ***	0.07 **	0.12 ***	−0.07 ***	−0.06
		(0.04)	(0.03)	(0.04)	(0.04)	(0.04)
Unemployment insurance		0.06 *	0.06 *	0.06 *	0.014 **	0.15 ***
		(0.03)	(0.03)	(0.03)	(0.04)	(0.04)
Hus education			−0.01	−0.02	0.02 ***	−0.02 ***
			(0.01)	(0.01)	(0.01)	(0.01)
Hus work			0.05 **	0.05 *	−0.11 *	0.07 ***
			(0.02)	(0.03)	(0.03)	(0.02)
Hus health			−0.01	−0.02	0.01	0.01
			(0.01)	(0.01)	(0.01)	(0.01)
Father			−0.03	−0.03 *	0.00	0.00
			(0.02)	(0.02)	(0.01)	(0.01)
Mother			0.06 ***	0.07 ***	−0.03	0.02
			(0.02)	(0.02)	(0.02)	(0.01)
Ln fincome			−0.01	−0.01	−0.00	−0.00
			(0.01)	(0.01)	(0.01)	(0.01)
Rich			−0.00	−0.01	−0.00	−0.00
			(0.01)	(0.01)	(0.00)	(0.00)
Regional fixed effect	YES	YES	NO	YES	NO	YES
*N*	925	925	1052	921	1896	2024

* <0.1, ** <0.05, *** <0.001.

## Data Availability

The data used in the paper come from the China Labor-force Dynamics Survey (CLDS), and the data is publicly available from https://www.mdpi.com/ethics.

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
