# Peer review of "Number of Children and Female Labor Participation in China"

_ijerph, 2022, doi:10.3390/ijerph19148641_

Round 1
Reviewer 1 Report
The authors corrected the text to the full satisfaction. The only drawback is the editorial problems: in this version of the text there are messages "Error! Reference source not found" and the footnotes are not standardized and the references are once directly in the text and other times in the indexes in square brackets, besides references to the surnames of the authors are first names (e.g. Alice & Masao instead of A. Nakamura, M. Nakamura). The text should be published - the topic of the article should be considered interesting (although it concerns only the situation in China). The literature review has been sufficiently in-depth and is now well done. The methodology was very well and logically described. The tables presented are selected adequately to the needs of understanding the discussed problem. It would be possible to broaden the context of the analysis and conclusions, however, and even in this form, the article provides a sufficiently new contribution to the scientific literature.
Author Response
Thanks for your suggestions!
Following your recommendations, we have made modifications in the following aspects after re-searching literature and sorting out relevant theoretical knowledge, and we have made uniform changes to the format and detail of the paper as required:
- For the introduction and literature review of this paper, we re-enriched the content of research motivation and research objectives. We elaborated the relationship between the number of children and the female labor force participation rate in detail, and further explained the theoretical significance of this study. At the same time, we have laid the groundwork for the possible theoretical transmission mechanism of this paper in the introduction, highlighting the theoretical mechanism and causal relationship behind the relationship. Supplement as follows: “In recent years, with the increasing problem of population aging, the problem of population structure has been paid more and more attention. Although the total population of our country continues to grow slowly and the rank of labor resources remains the first place worldwide, at the same time, it was 13.50% high that the proportion of our country’s population aged 65 years and older has reached, and the elderly population ushered in a new total.”
- In the literature review and research hypothesis part, we have modified some detailed issues, such as authors’ names: Alice & Masao instead of A. Nakamura, M. Nakamura. We have made the following modifications: “Angrist & Evans (1988) , Alice & Masao (1992) believe that the increase of the number of children will remarkably reduce the probability of women’s labor participation.”
- As for the details and format of the paper, we have checked and corrected the full text. We have adjusted the format of the footnotes appearing in the paper, and revised the paragraph symbols to ensure the accuracy of the overall format. Based on the comments of reviewers, we have standardized the footnotes and standardized the format to make them look better. Thank you for correcting the details of the paper !
Reviewer 2 Report
While the manuscript has improved in some respects and it leads to interesting conclusions, there remain aspects that require further work before it can be considered ready for publication. Among these are the following:
-The theoretical mechanisms underlying the expected or observed relations are still often not entirely clear/sufficiently spelled out. Greater specification, detail, and clarity is needed in this regard. No causal mechanism should be taken for granted or explained vaguely. Any proposed causal mechanism/relation should be thoroughly and clearly explained from the beginning (especially in the introduction & literature review sections)
-The research questions presented at the end of the introduction should be more clearly related to the theoretical mechanisms expected to be at play or, at least, more explicitly clearly rooted in earlier research/existing knowledge on the topic.
-While the methodological design and execution are appropiate, the manuscript lacks a clear theoretical "red thread" and a more systematic organization structured around the theoretical mechanisms expected, how these are tested, and the results obtained. Theoretical arguments and reasoning appear often "ad hoc" in the results section and have not been sufficiently developed in the introductory and literature review sections. This makes the manuscript confusing to read and follow. I would advise the authors to place focus from the beginning on "1) The importance of the question that is going to be analyzed (i.e. "why is it important to analyze the factors that affect female labor market participation in China"? 2) What are the theoretical factors that would lead us to expect the number of children to affect female labor market participation in China? In which directions could they work? 3) Exactly in which ways could regional heterogeneity affect results?". These aspects should subsequently guide in a systematic way the organization of the different sections and be underscored in the conclusions/discussion section as well. The manuscript would also benefit from a clearer, more to the point presentation of results (centered around objective findings, and not so much around theoretical reasoning, which should rather be in focus in the introductory, literature review an concluding sections).
-The manuscript also needs relatively extensive language editing.
Author Response
Thanks for your suggestions!
For the mechanism analysis and key elaboration part of the paper you mentioned, we have made a comprehensive revision and improvement according to your revision comments. We have made modifications in the following aspects after re-searching literature and sorting out relevant theoretical knowledge, and we have made uniform changes to the format and detail of the paper as required:
- In the part of introduction, we summarizes the differences in labor employment patterns between China and other countries, and analyzes various factors that affect the female labor force participation rate. The specific modifications are as follows: “At the same time, existing experience has proved that the female labor force participation rate and its influencing factors are different in China and other countries. One of the important reasons is that China has ranked first for its workforce participation rate is that the women’s labor-force participation rate is much higher than the world average.” ; “What factors would make the expected number of children affect the participation of Chinese women in the labor market? Various real-world analyses show that analyzing the factors that affect China’s female labor market is of great significance to China’s population structure improvement and social development.”
- At the end of the introduction, in order to highlight the research contribution of this paper, we add a cross-examination of the main research question of the paper, so as to clearly describe the idea of the research question of this paper: “More importantly, what is the transmission mechanism and theoretical influence mechanism between the number of children and the female labor force participation rate studied in this paper? And does this theoretical transmission mechanism match our expectations or reality?”.
- In accordance with the conclusion of benchmark regression, we have sorted out the conclusions and policy recommendations, and explained the conclusions drawn in this paper one by one for the conclusions drawn from benchmark regression and heterogeneity testing, and put forward targeted policy recommendations. For example, we added the policy which is “Raising women’s labor force participation might offset ongoing declines in male participation, and doing that will require support for women who have relatively large numbers of children.”
- In the literature review section, in order to further reflect the causal mechanism or the role of variables in the transmission mechanism of this article, we give a general description of the factors that affect female labor force participation, so that readers can grasp the logic more clearly. The supplementary content is as follows: “Due to factors such as age, gender, educational development, income growth, and industrial structure, the issue of female labor force participation cannot be ignored, and its role in social and economic development is self-evident. For women, their labor participation in decision-making is affected by marriage, family, childbirth and other factors, which can not be ignored.”
Thank you for suggestions and corrections again !
Reviewer 3 Report
This paper aims to promote understanding of the forces affecting women’s labor force participation patterns in China, especially in the context of the aging of the population and the loosening of fertility policy. This is an important and worthwhile task. The authors use cross-sectional data from the 2016 China Labor-Force Dynamic Survey to examine this issue, focusing on the impact of the number of children born on the likelihood of a women being a labor force participant.
Their results indicate a quadratic effect of children on labor force participation: the odds of participation rise with the number of children at low levels of childbearing but fall at high levels of childbearing. This is interpreted as reflecting an “income effect” of childbearing at lower levels (women work more to cover the cost of raising kids) and a “substitution effect” at high levels (women reduce work in order to take care of their many children). These patterns hold more strongly in urban and developed (eastern and central) places. Rural places, and the less-developed west, are marked by little influence of fertility on labor force participation.
These are interesting and important questions, and I think the regional differences found here are especially intriguing. However, I have a number of concerns about the framing of the question and the execution and presentation of the empirical work.
With regard to the framing of the question:
The introduction and literature review are a bit hard to follow, so it is difficult to get a clear sense of the central question. Some of this may just be language and translation issues. Some of the problem is a matter of organization. I would recommend revising this section to more clearly highlight (1) what we think we know about the relationship between fertility and women’s labor force participation in general, (2) reasons for believing that these patterns might be different in China, and (3) any existing evidence that China is distinct in these patterns. Much of this information is provided, but not in a very organized way, and not in language that is very clear to readers of English.
With regard to the execution and presentation of the empirical work:
Table 1 provides a description of the variables in the analysis. The presentation seems to confuse “explanatory” and “explained” variables. Labor force participation is the “dependent variable” in this analysis, so I would consider that the “explained” variable (though “dependent” would be a better term). Then number of children and age would be “explanatory” variables.
Based on the description in Table 1, it’s not clear to me that the dependent variable is actually “labor force participation.” Labor force participants include both those working and those seeking work (even if they haven’t found it). Table 1 seems to say that labor force participation here is defined as “being employed.” That’s different from the usual use of this term in labor economics. This point should be clarified.
A number of job characteristics are included as explanatory or control variables in the analysis. This strikes me as a little strange: in what sense can the characteristics of a particular job be considered to be determinants of one’s choice to be a labor force participant – that is, the choice of whether or not to seek work? How are these job characteristics (for instance, “work unit” and “full-time”) measured for individuals who are NOT labor force participants or NOT employed? (There must be some non-participants or unemployed people in the data set in order to estimate these models, right?)
How should we interpret the coefficients on “unit” in the regressions? “Unit” just appears to be a categorization of types of work places – public, state owned, foreign capital, individual – but it seems that it enters ordinally (from 1 to 4) in the regression? In the discussion of table 3, we learn that “The marginal effect of workplace type [unit, I think] is positive, showing that the more flexible the workplace type, the higher the female labor-force participation.” So is the categorization of units from less (1) to more (4) flexible types? (Is that explained somewhere?) Why not just use a separate dummy variable for each type of “unit”?
It’s also unclear what the “unemployment insurance” and “maternity insurance” variables are measuring. This should be explained for those who are not familiar with the Chinese context.
The set of included variables varies in surprising ways across the models. In Table 3, we get only a quadratic version of age (OLD2), and no linear version. This occurs again in Table 5, Table 6, Table 8, Table 9, and Table 10. The discussion of these models generally includes consideration of “U-shaped” patterns of participation with respect to age, so perhaps the omission of the linear form of age is just an editing error?
Both linear and quadratic forms of age are included in Table 4. The discussion of the results, on Page 11, describes an “inverted U shaped pattern” of impact of age on participation, though what is described sounds like a regular(not inverted) U shaped pattern (high participation at young and old ages, and lower participation in prime child-rearing years), which is what I’d expect. Is the discussion on page 11 consistent with the coefficients in Table 4?
Several “robustness tests” are performed, including the estimation of a logit model and a re-estimation excluding “exteme values.” I don’t think the logit estimation tells us much (it would be very surprising if the probit and logit results differed in any important way).
I’m a little confused about how the “extreme values” check is done. We are told that “the number of children was subjected to 1% and 3% tailoring,” which I take to mean that observations in the top or bottom 1% (or 3%) of the fertility distribution are dropped. But the number of children ranges just from 1 to 7. If more than 1 or 3 percent of the observations have just 1 child, or if more than 1 or 3 percent have 7 kids, how are the excluded observations chosen? In addition, on page 7 we are told that “Women have an average of 1.76 children, of which 2,767 have 1-2 children, accounting for 89.2%; 335 women have three children, accounting for 10.8%.” That’s all 3102 observations – so, again, how are “extreme values” trimmed out of this distribution? And how is this description consistent with the maximum number of kids being 7 in Table 2? Maybe 10.8% of women in the sample have 3 or more kids?
I’m also confused by the presentation of the instrumental variable results. See Table 7. The table lists Child Num as an explanatory variable in the “first stage” regression, but the discussion suggests that Child Num is the dependent variable in this regression, and that age at first birth is the main explanatory variable. Am I misinterpreting something? Why are only quadratic child terms (not linear) included in the full models (columns 2, 4, and 5)? Why are the parameters in the “Child Num 2” (quadratic) rows discussed as if they are linear effects of childbearing on labor force participation? For instance, on page 16, we are told that “The results show that after treating 'age at first birth' as an instrumental variable and adding regional fixed effects, having an additional child increases female labor force participation by 8.2%,…” but 8.2% here seems to refer to the entry in the “Child Num 2” row in model 5?
Finally, the recommendations in the conclusion - promoting gender equality, reducing discrimination, etc. – are good things, but they don’t follow in any obvious way from this analysis. I guess the point is that raising women’s labor force participation might offset ongoing declines in male participation, and doing that will require support for women who have relatively large numbers of children (given changes in fertility policy in China). I think these recommendations could be connected more directly to the results of the empirical work.
Author Response
Thanks for your suggestions and opinions !
Thanks to the reviewer for pointing out the empirical problems of the paper and the details of the format, we have made a comprehensive revision and improvement according to your revision comments. We have made modifications in the following aspects after re-searching literature and sorting out relevant theoretical knowledge, and we have made uniform changes to the format and detail of the paper as required:
- As for the measurement of independent variables, we have explained in detail in the relevant positions in the text: “Since the independent variable in this study is the number of children, the quadratic regression results of the number of children show a special phenomenon, which shows that the number of children and the female labor force participation rate are in an “inverted U” shape, which indicates that the number of children is closely related to the female labor force. Participation is not a simple linear relationship, so the follow-up research in this paper does not directly use linear variables for regression, which may lead to biased conclusions, but uses quadratic variables to verify conclusions and explain problems.”
- Regarding the two variables at the social security level you mentioned which are “unemployment insurance” and “maternity insurance”, we have explained in detail in the paper, and specifically explained the content and significance of the two variables measured. The specific instructions are as follows: “With the continuous progress and development of social security, the employment situation of female labor force is also affected by social factors. As a typical measure of social security, insurance is gradually accepted and popularized by social groups. This paper selects two social security variables which are unemployment insurance and maternity insurance to measure the impact of social factors on female labor force participation.”
- In the Data Selection part, in accordance with the requirements of the reviewers, we have revised the descriptions of important variables in the text, changed “explaining variables” to “independent variables”, changed “explained variables” to “dependent variables”, and carried out in the table unified changes.
- Regarding the selection of independent variables in the regression of instrumental variables, we conducted a detailed analysis and discussion, explaining its role in the regression and why the quadratic of the number of children was selected as the main regression variable. The specific instructions are as follows: “Since the independent variable in this study is the number of children, the quadratic regression results of the number of children show a special phenomenon, which shows that the number of children and the female labor force participation rate are in an inverted "U" shape, which indicates that the number of children is closely related to the female labor force. Participation is not a simple linear relationship, so the follow-up research in this paper does not directly use linear variables for regression, which may lead to biased conclusions, but uses quadratic variables to verify conclusions and explain problems.”
- As for the details and format of the paper, we have checked and corrected the full text. We have adjusted the format of the footnotes appearing in the paper, and revised the paragraph symbols to ensure the accuracy of the overall format. Based on the comments of reviewers, we have standardized the footnotes and standardized the format to make them look better. Thank you for correcting the details of the paper !
Thank you for suggestions and corrections sincerely !
Round 2
Reviewer 3 Report
Please see my comments in the attached file.

Author Response
Thanks for your suggestions and opinions !
Thanks to the reviewer for pointing out the empirical problems of the paper and the details of the format, we have made a comprehensive revision and improvement according to your revision comments. We have made modifications in the following aspects after re-searching literature, and we have made uniform changes to the format and detail of the paper as required:
- As for the measurement of independent variables, we have explained in detail in the relevant positions in the text: “Labor participation refers to the entire population of a certain age, with labor ability and employment requirements, engaged in a certain occupational labor.” This corresponds to the meaning of the variables in Table 1. The labor situation corresponds to the state of “employment”, not the labor force in the economic sense. In this part, we consider that the group who is looking for a job is not included in the scope of “labor force”. And according to the definition of labor force participation in the existing literature, we believe that the purpose of seeking labor participation is to achieve the state of being “employed”. We have further supplemented this: “Therefore, the focus of this paper is on how the number of children affects whether women find work, rather than whether women are looking for work.”
For the question of other job characteristic variables, we understand that if it does not belong to labor participants, it does not belong to the full-time labor force in a certain sense. At the same time, if a worker signs a labor contract or agreement with the employing unit and determines that he belongs to the employed person, then he has the exact work unit, although he may not belong to the state of being employed every day. Thanks for the reviewers for raising the issue of defining variable definitions !
- In response to the variable definition problem of “unemployment insurance” and “maternity insurance” proposed by experts, we understand that according to the system and policy requirements of modern social security, it is not purchased after the laborer’s career, but after the laborer decides to find employment. Buy this kind of insurance for yourself or your family at the time to truly achieve the purpose of unemployment benefits. To understand it from another angle, if you buy unemployment insurance only after you lose your job, then there will be a large degree of adverse selection and moral hazard problems. Therefore, both “unemployment insurance” and “maternity insurance” in this paper belong to insurance directly purchased by individuals or families. I hope my answer can successfully answer your concerns.
- In the Data Selection part, we measure the variable age in two ways, based on the practice used in many literatures. First, the linear form of age is the most common and most well-accepted, and has been measured in much of the literature, so we add it. The "quadratic of age" is determined according to the actual influence of the paper, because the influence of the number of children on female labor force participation is not linear, and this variable is the main explanatory variable to explain the conclusions of the paper. Therefore, in Table 5, Table 6, Table 8, Table 9, and Table 10 involving model regression later in the paper, only the “quadratic age” variable appears, which is sufficient to explain the main research problem. I hope this answer could explain your doubts about the existence of variables.
- In the endogenous test part, we apply 2SLS (2 stages Least Square) regression analysis to explain the endogenous problems of the model. The essence of 2SLS is to divide the endogenous explanatory variables into two parts, that is, the exogenous change part caused by the instrumental variables, and the other parts related to the disturbance term; then, the exogenous part of the explained variable pair is regressed. , so as to meet the requirements of OLS pre-determined variables and obtain a consistent estimator. The reason why the first column uses the primary term of the number of children as the explained variable is that although we focus on the relationship between the quadratic number of children and female labor force participation, the endogeneity test is relatively independent compared to the benchmark regression. part, it is still necessary to regress the power of the number of children to take it into account.
- As for the details and format of the paper, We numbered references in the order in which they appear in the text (including citations in tables and figures) and listed them separately at the end of the manuscript as requested by reviewers. In the text, we have placed the reference number in square brackets [] and before punctuation as required. At the same time, we have edited the highlighted part according to our actual situation at the end of the paper.
Thank you for suggestions and corrections sincerely !
This manuscript is a resubmission of an earlier submission. The following is a list of the peer review reports and author responses from that submission.
Round 1
Reviewer 1 Report
This paper attempts to examine the relationship between the number of children and women’s labor force participation. Using data from the 2016 China Labor-force Dynamic Survey, the authors find that there exists an inverted U-shaped relationship between the number of children and the rate of women’s labor force participation. Although the paper has a clear research objective and provides interesting empirical results and useful policy implications, I find that there are several areas of the paper that need to be improved.
1) The authors offers a theoretical basis for understanding the relationship between the two key variables, employing income and substitution effects. Then readers expect to have an interpretation of the main empirical results as well based on the aforementioned theoretical channels, which is absent in the paper. To be more specific, what are the theoretical reasons behind the inverse U-shaped relationship, which indicates the income effect is stronger when the number of children is small but the substitution effect outweighs the income effects after some threshold value?
2) In terms of the specification of the model, potentially one important omitted variable is person-specific unobserved heterogeneity. Given that the China Labor-force Dynamics Survey has the rotating panel structure. panel model can be employed to control for unobserved heterogeneity and it is necessary to check the sensitivity of the main results after controlling for the household fixed effects.
3) The sample includes women aged between 20 and 51. However, women in twenties and early thirties may have to deal with the joint decision problem regarding labor market participation and childbirth. Because of the potential simultaneity or reverse causality, it is worth considering robustness exercise using the subsample of women between 35 and 51.
4) In the introduction of the paper, it is stated that the female labor force participation rate in China is around 60% as of 2019. However, the descriptive statistics show that the corresponding figure is 92.6% in the sample. The large discrepancy requires some justification.
5) Is there any reason why only the age square term is included (but not the age term) in the model in Table 7?
6) There are many grammatical errors and typos including, but not limited to, the following:
“ More and more women choose have fewer children or not to have children”
“The influence way of raising children on women's labor participation is not single, and there are income effect and substitution effect”
“which is because that women in China have a stronger choice of labor force participation”
“they may be gradually separated from the workplace in psychology and behavior”
Reviewer 2 Report
Many thanks for giving me the opportunity to review this manuscript.
The paper deals with a scientifically and socially relevant topic (the relation between the number of children and women's labor market participation in China). It is generally well and clearly written (although it needs English language editing), methodologically very thorough and sound, and the conclusions are supported by the results. There are nonetheless certain issues that I would recommend addressing before it can be considered ready for publication:
-In the introduction, although adequate background information is provided, it would be advisable to state more explicitly the scientific and societal relevance of the contribution (e.g. which kind of gap in earlier research does the manuscript fill? why is it societally relevant to answer the analyzed questions in the Chinese context?).
-On p.2 line 79, it should be specified that the type of heterogeneity analyzed is regional heterogeneity (and the authors should briefly mention already in the introduction why it is important to take it into account)
-In the literature review section, it would be advisable to specify on p.3, lines 107-108 in which ways the economic system transition has proven to be an important factor affecting women's labor participation rate in China.
-In the literature review section (p.3, lines 110 and onwards), it is stated: "However, due to the differences between estimation methods and data processing, it is unable to obtain a unanimous conclusion". In which ways do the authors expect that their contribution will offer more robust results than previous analyses of the phenomenon? Clarifying this would also highlight the scientific, substantive and methodological contribution of the manuscript.
The literature review offers many (and relevant) examples of empirical contributions dealing with the analyzed topic and related issues in China and other countries. Nevertheless, a clearer theoretical framework is neeeded. Why is it expected that the number of children should affect women's employment participation in the ways hypothesized? Which kind of mechanisms could be at play? Is there potential endogeneity/reverse causality? (this is dealt with further along in the analysis, but it should already be discussed in the theory section, and also in the methods section). Could there be some sort of selection bias? (e.g. that those Chinese women who are more prone to show a relatively high employment participation also are more prone to have a given number of children?) To describe possible theoretical mechanisms/relations more clearly, I would suggest building on the economic literature on the topic (e.g. the seminal works of Becker, Mincer, Angrist and Evans - which are already mentioned -, and later contributions working on the same tradition).
A clearer, more ordered and systematic theoretical discussion is also required regarding the Chinese case. Which relations could be theoretically expected between women's labor market participation and number of children in China? How could the countries' specific fertility policies have impacted these relations? Which regional differences might we expect and why? What theoretical reasons are there to expect the non-linear relation between the dependent and independent variable analyzed and observed? The mechanisms based on "income and substitution effects" should be explained more clearly in the theoretical section (they come forward more clearly in the results section, but it is important to present them in a clearer, more systematic fashion in the theoretical section as well). This would allow readers who are unacquainted with income and substitution effects to understand more easily the rationale for the research hypothesis formulated. In short, a bit more clarity regarding the explanations of such mechanisms would be advisable.
-On p. 4 (line 1987 and onwards) it is not explained why women, in China, would need to "bear more family responsibilities, consume more energy in the family, and have closer ties with the family" compared with men. Is it due to cultural reasons/traditions/normative social pressure? It would be advisable to clarify this.
-On p. 5, line 215, it should be specified how women's "working status" has been operationalized (as a dychotomous variable. It becomes evident later in the manuscript, but it should be clearly specified already at this point)
-Does the control variable "health" make reference to self-perceived health status? It should be clarified.
-There is a typo in table 1. The variable name corresponding to the "name in regression" "mother" should be "whether the mother is alive", instead of "whether the father is alive".
-How is the extent of family wealth measured? Please specify the variable's operationalisation more clearly. What does "getting richer" mean in this context? (table 1). How is the original variable in the dataset constructed?
-Regarding the model setting, why is a traditional OLS model first used if the dependent variable is a dychotomous one? Why not go directly to a logit or probit model? It would be advisable to justify the choice of using the two types of models. Is it for robustness testing purposes? Is it because the authors intended to estimate an effect through OLS and then a probability through the probit models?
-What is meant by "the working status of women" (a set of control variables, X2) on page 7? If the term makes reference to the women's employment characteristics (as seems to be the case), I would suggest shifting to this denomination instead (the women's employment characteristics), as the term working status can be confusing (it usually refers to whether the individual is working/employed or not).
-I would recommend briefly explaining in section 3 how and why regional fixed effects are included and how regional heterogeneity and the possibilit of endogeneity are dealt with. Although these aspects do come forward in the results section (section 4), it is necessary to explain them beforehand (in the methods section).
-On p. 10, lines 391 and onwards, it is not very clear what is meant by "Spouse's participation in the labor market increases women's labor force participation by 0.51, which is because women in China have a stronger choice of labor force participatio and spouse's particiàtion in the labor market motivates women to pursue their own career achievements". Why should this be so? I would recommend clarifying this and making a stronger, theoretically reasoned argument.
The conclusions should be more explicitly connected to the theory section and the theoretical arguments made.
Reviewer 3 Report
The motivation and contribution of the manuscript are too marginal. Without reading it, people can expect all three results in the manuscript.
I would like to encourage the authors to identify the driving forces of the three results.
Reviewer 4 Report
The article should be assessed as interesting and well prepared in terms of content and methodology. However, a few amendments could raise its value even more.
The authors analyzed the situation in China and mainly refer to the literature in this field. It seems, however, that it would be important to refer to the broader context in order, firstly, to see if the situation in China is exceptional, and secondly - what the research problem may look like in other countries and how it has been described. It would also enrich the literature review, which is relatively poor. Moreover, it would be worth indicating which of the research conclusions indicated in the text concern China and which of other countries. One of such places is the statement that: "Zhang (2020), Zhang & Gu (2020) also believe that fertility is one of the key factors leading to the depreciation of human capital of urban women, and fertility has a greater negative impact on the employment of urban women and highly educated women. " - it is not known whether it concerns the situation in China or the research covered a wider population.
The considerations for urban-rural heterogeneity, and especially the results for urban women, are not sufficiently clear. It would be necessary to correct this passage, because now the interpretation of the results is not understood and explained appropriately and transparently.
It would also be worth clarifying the fragment: "The results show that after treating 'age at first birth' as an instrumental variable and adding regional fixed effects, having an additional child increases female labor force participation by 8.2%, but is not significant, possibly due to the influence of the one-child policy and changes in fertility attitudes, with a larger sample of only children in the sample and an average number of children of 1.76, which affects the significance of the number of children coefficient. coefficient has a positive sign, which is consistent with the above regression results. This indicates that a certain increase in the number of children does not reduce female labor force participation. On the contrary, it will facilitate female labor market participation. ". Especially the conclusion from the last sentence is not obvious.
The authors state that "With China's economic development and the increase of family income, residents with higher family income will pay more attention to the quality of child rearing and reduce the number of children." - it seems obvious that it is necessary to refer to the theory of G.S. Becker.
In the sentence: "The average health status in the sample is 2.3, and the per capita health status is considerable." the phrase referring to the value 2.3 is awkward and illegible. It is not known what exactly 2.3 means, it is rather the value used to code the response in the study, not a measure of health status, so it needs to be reformulated.
Technical note - there are footnotes in the text that do not know what they are referring to, because they are not below the text (it is not about footnotes in circles), there are also references in square brackets (again - it is not known what they refer to) and in in the text (in the literature review) the following messages appear: "Error! Reference source not found. Error! Reference source not found." It is necessary to check the entire article from this point of view and adapt the text to the reference system used in the journal.
Author Response
Reply
Thanks for your suggestions!
Following the recommendations of reviewers, we have made modifications in the following aspects after re-searching literature and sorting out relevant theoretical knowledge, and we have made uniform changes to the format of the paper as required:
- For the introduction and research implications of this paper, we re-enriched the content of research motivation and research objectives. At the beginning of the paragraph, we added a description of the significance of studying the number of children and female labor participation to highlight the purpose of this study.
- For the inconsistency between the data used in the paper and the national statistics, we have revised and explained in the introduction. Due to the statistical scope and data processing reasons of the statistical data, the data of CLDS 2016 is quite different from the statistical data of 2019. The specific changes are as follows: “The data used in this study is CLDS 2016, and the year is quite different from 2019, so the female labor force participation rate is relatively high. In addition, the statistical caliber of this data includes more urban data from 29 provinces in China. From the perspective of the accuracy of the empirical results, sparsely populated remote areas are excluded. Therefore, the average value of this indicator is higher than the statistical data in 2019”.
- At the end of the introduction, in order to highlight the research contribution of this paper, we add a cross-examination of the main research question of the paper, so as to clearly describe the idea of the research question of this paper: “Specifically, why is there strong regional heterogeneity in the question between the number of children and the female labor force participation rate in China? What are the theoretical factors inherent in it?”.
- In the literature review and research hypothesis part, we have added an elaboration of the research background of the paper, explained the economic and social background of the study, and clarified the field and significance of the research: “With economic and social development, people’s ideological level has been continuously improved, modern society has paid more attention to women, and women’s status has been continuously improved. Increasingly studies are also exploring the influence mechanism and relationship between the number of children and the female labor force participation rate in the family relationship”.
- In the literature review part, in accordance with the requirements of the reviewers, we describe the research on women's labor participation in China, and compare foreign research with Chinese research, so as to highlight China's current research innovations and differences. The specific modifications are as follows: “The impact of fertility rate on women’s labor supply behavior is one of the important topics in the field of labor economics. Although there have been a lot of empirical studies on women’s labor supply behavior in the United States and western developed countries, research on women’s labor supply behavior in developing countries is still very limited. The decline in the female fertility rate in my country may lead to an increase in the female labor force participation rate, change the female employment structure, and may also narrow the gender wage gap.”
- According to the suggestion, we have explained the effects of female labor participation of different groups under different conditions. Combined with traditional Chinese thought and consciousness, we explain the reasons as follows: “This is because the concept of a matriarchal society is deeply rooted in the hearts of the people in China. It is always believed that in a family, the role played by women is often the most critical. It will play a more subtle and irreplaceable role in the construction and maintenance of a home. At the same time, traditional cultural factors also believe that women often use softness to overcome rigidity, and to a large extent play a more firm backing role than men”.
- In accordance with the suggestions of reviewers, we have further expanded the literature research on women's labor participation, and added relevant foreign research to broaden the research perspective of this paper: “The main difficulty in studying the effects of fertility on women’s labor supply behavior is that women’s reproductive decisions and labor supply decisions may be made at the same time and affect each other. Women who tend to participate in formal employment may also tend to have fewer children, and employment status may also affect subsequent reproductive behavior. In addition, there are other unobtainable parental characteristics such as women and their spouses, which affect both reproductive decision-making and labor supply. Existing studies have used the instrumental variable method to solve the endogeneity problem in identification. Das et al. (2003) and Ebenstein (2010) used twins as an instrumental variable for the number of children”.
- In the part of 2.2 Data Selection, we have detailed the variables that need to be explained in strict accordance with the recommendations, explaining how household wealth and personal wealth are measured and the types of variables, as follows: “Among them, the family wealth is measured by the variables Extent of family wealth and Family annual income. In particular, the Extent of family wealth is an ordered discrete variable, and it is a variable that is positively measured. From 0 to 10, it means that the family wealth is more It is a more objective and comprehensive measure to use the size of the numbers to measure the comprehensive wealth of the family more intuitively. Family annual income measures the material wealth of the family from the dimension of family income.”
- In Table 1, we changed the “father” to “mother”, and checked the other variables in the table one by one. Thanks to the reviewers for their careful review!
- In the variable description part, in order to make the statistical results of the variable clearer, we have made a statistical description of the health status and explained what its mean value represents: “The average health status in the sample is 2.3, and it can be seen from the variable assignments in Table 1 that the value of the health status variable is measured inversely, that is, the smaller the value of the variable, the better the health status. Therefore, the mean value of health status in the sample is 2.3, and the per capita health status is considerable.”
- In the Model Setting part, we added the reason for choosing the OLS model and the Probit model, and the meaning of the random error term that may exist in this paper is explained as required: “Since the Least Squares (OLS) is the most traditional and common method in regression, it is suitable for various types of variable regression. Therefore, this study firstly used OLS to perform regression analysis of the model.”; “In addition, the Probit model is suitable for variables whose type is ordinal and discrete. Therefore, this paper selects the ordinal Probit model for benchmark regression and robustness test regression.”
- In the empirical analysis part, we explained the detailed mechanism of the fixed effect model and the reasons for adding it as required, and clarified the necessity of adding fixed effects. The details are as follows: “The reason for adding fixed effects is that the relationship between the number of children and the female labor force participation rate is largely affected by regional factors. For example, there are differences in the number of women and their employment development in the western and eastern regions, and because the economic development of the two regions is quite different, the research on the effect of regional factors on the female labor force participation rate cannot be ignored. Based on this analysis, this study added a regional fixed effect to the model regression. Specifically, this paper adopts the method of fixing each province to deal with fixed effects.”
- In the footnotes to the regression results, we updated and further clarified the description of the significance of the model results, explaining the meaning of the values in parentheses in the table: “This section describes the significance of the regression results in the table. Standard deviations in brackets, *** indicates significant at the 1% level, ** indicates significant at the 5% level, * indicates significant at the 10% level. The following table is the same as above.
- Behind the regression table, we explained the relationship between women’s labor participation and their spouse’s labor participation as required, and explained the social reasons for this result: “Women’s participation in the labor market increases 0.51 when her spouse labor force participation increasing one unit, which is because that with the development of the economy in modern society, women’s ideological and viewpoints are gradually breaking the shackles. They believe that women should not only be husbands and children, but should pursue higher social status and highlight their social values. The experience that men get at work is more attractive to women’s desire to work, so the above regression results appear.”
- According to the requirements, we have rearranged the conclusions of this paper according to the logic of the full text, and presented the conclusions drawn by each part of the research in sub-items: “(1) The relationship between the number of children and female labor and participation rates is not a simple linear relationship, but a nonlinear relationship with an inflection point. An increase in the number of children has a robust “inverted U-pattern” relationship with female labor force participation, which holds true after controlling for female personal characteristics, work characteristics, family characteristics and the addition of provincial fixed effects. (2) Further research found that higher prior-year earnings, unemployment insurance, and spousal labor status also significantly increased the likelihood of female labor force participation in China. (3) With the development of economy and society, the impact of economic structure and development speed on the female labor force participation rate cannot be ignored, and external factors such as regions will also affect the conclusions of this paper. In terms of heterogeneity, an increase in the number of children has a more significant “inverted U-pattern” relationship with female labor force participation in eastern and central China, and an increase in the number of children has a more significant “inverted U-pattern” relationship with female labor force participation in urban areas than in rural areas. The existence of provincial fixed effects can also explain the influence of regions on the relationship between the number of children and the female labor force participation rate.
- Based on the content and format of the literature review in the main text, we have restructured the references and added two English-language papers. At the same time, We checked the format and content of this paper, and corrected the wrong format or the incomprehensible statement. Thank you for suggestions and corrections again!